# Sparse Iso-FLOP Transformations for Maximizing Training Efficiency

**Vithursan Thangarasa**[*], **Shreyas Saxena**[*], **Abhay Gupta**, **Sean Lie**
Cerebras Systems
{vithu, shreyas.saxena, abhay, sean}@cerebras.net

## Abstract

Recent works have explored the use of weight sparsity to improve the training efficiency (test accuracy w.r.t training FLOPs) of deep neural networks (DNNs). These works aim to reduce training FLOPs but training with sparse weights often leads to accuracy loss or requires longer training schedules, making the resulting training efficiency less clear. In contrast, we focus on using sparsity to increase accuracy while using the same FLOPS as the dense model and show training efficiency gains through higher accuracy. In this work, we introduce Sparse-IFT, a family of Sparse Iso-FLOP Transformations which are used as drop-in replacements for dense layers to improve their representational capacity and FLOP efficiency. Each transformation is parameterized by a single hyperparameter (sparsity level) and provides a larger search space to find optimal sparse masks. Without changing any training hyperparameters, replacing dense layers with Sparse-IFT leads to significant improvements across computer vision and natural language processing tasks, including ResNet-18 on ImageNet (+3.5%) and GPT-3 Small on WikiText-103 (-0.4 PPL), both matching larger dense model variants that use 2x or more FLOPs. To our knowledge, this is the first work to demonstrate the use of sparsity for improving the accuracy of dense models via a simple set of sparse transformations. Code is available at: https://github.com/CerebrasResearch/Sparse-IFT.

## 1 Introduction

Increases in model size and training data have led to many breakthroughs in deep learning (e.g., AlexNet [41], ResNet [30], Transformers [89], GPT [69, 70], AlphaGo [77], etc.). Consequently, computational and memory demands for training and deploying deep neural networks (DNNs) have surged dramatically. To enable the deployment of large models, multiple techniques (e.g., distillation [32], quantization [28], pruning [29]) have been introduced to reduce inference FLOPs and memory requirements. While these techniques improve inference efficiency (test accuracy w.r.t inference FLOPs), the associated training costs are still prohibitive. In this work, we focus on improving the training efficiency (test-accuracy w.r.t training FLOPs) of DNNs.

Recent works [19, 37] have explored using weight sparsity to reduce the FLOPs spent in training. Frankle and Carbin [20] demonstrate that sparse subnetworks (termed "lottery tickets") exist at initialization and can be trained to match the accuracy of their original dense network. Inspired by this result, various dynamic sparse training (DST) methods [19, 37, 52, 58] attempt to find optimal sparse subnetworks within a training run. While these methods primarily aim to improve training efficiency by reaching dense accuracy with fewer FLOPs, they often perform worse than their dense baselines or rely on longer training schedules (up to 2-5× training iterations) to close the gap [51, 83, 94].

---

[*]Equal Contribution.

Workshop on Advancing Neural Network Training at 37th Conference on Neural Information Processing Systems (WANT@NeurIPS 2023).

As a result, these techniques can sometimes even require more FLOPs than training the dense model [19, 37, 58]. Our aim is to highlight our unique contribution in utilizing sparsity to enhance standard dense model accuracy, distinguishing our work from previous research. While past studies focused on pruning techniques to improve accuracy of pre-trained dense models [29, 56, 63], our innovation lies in demonstrating sparsity's impact on accuracy when training from scratch within the same training FLOP budget as dense models. Specifically, we introduce a family of Sparse Iso-FLOP Transformations (Sparse-IFT) that can be used as drop-in replacements for dense layers in DNNs.

These transformations increase the representational capacity of layers and facilitate the discovery of optimal sparse subnetworks without changing the layer's underlying training and inference FLOPs (i.e., Iso-FLOP). For example, making a layer wider but sparser increases dimensionality while still maintaining FLOPs due to sparsity. All Sparse-IFT members are parameterized by a single hyperparameter, the sparsity level. Figure 1 summarizes the ImageNet performance with ResNet models, where our Sparse Wide IFT variants significantly increase the accuracy of matching Iso-FLOP dense models. In particular, Sparse Wide ResNet-18 at 90% sparsity improves the top-1 accuracy from 70.9% to 74.4% (+3.5%), and outperforms a dense ResNet-34 (74.2%) while using 2x fewer FLOPs. We emphasize that these gains were obtained by replacing dense layers with transformations from the Sparse-IFT family

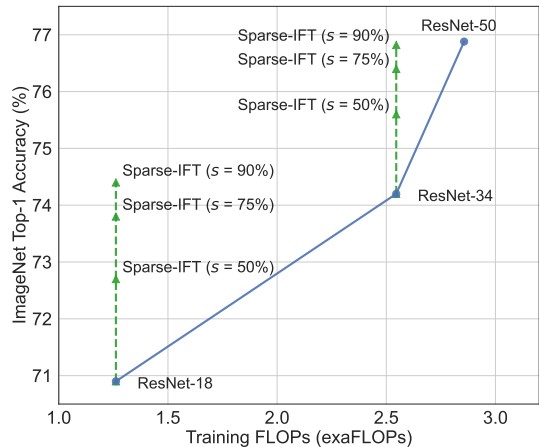

Figure 1: Accuracy vs. Training FLOPs for different variants of ResNet on ImageNet. Sparse-IFT provides significant accuracy gains across different models and sparsity levels while using the same FLOP budget as its dense counterpart.

and required no changes to training hyperparameters. The main contributions of our work are:

1. We introduce Sparse Iso-FLOP Transformations (Sparse-IFTs), a family of techniques aimed at enhancing DNN training efficiency. These transformations boost accuracy while maintaining a constant FLOP count. Sparse-IFTs are characterized by a *single hyperparameter, sparsity level*, and can be seamlessly used as drop-in replacements for dense layers.

2. In the CV domain, using Sparse-IFT increases the top-1 accuracy of ResNet-18 and ResNet-34 by 3.5% and 2.6% respectively on ImageNet. Finetuning these pre-trained models for object detection (MS COCO) and segmentation (CityScapes) leads to an improvement of 5.2% mAP and 2.4% mIoU, respectively.

3. In the NLP domain, using Sparse-IFT with GPT-3 Small leads to a 0.4 perplexity improvement on the WikiText-103 language modeling task, and matches the PPL of a dense GPT-3 Medium while using 2.4x fewer training FLOPs.

## 2   Method

In this section, we present our method to improve training efficiency. We first explain our intuition and hypotheses, followed by our methodology.

**Training with Dense Matrices is FLOP Inefficient**   Prior research indicates that modern DNNs are overparameterized, and they exhibit sparsity in both features and weights across layers. The Lottery Ticket Hypothesis (LTH) [20] demonstrates that sparse DNNs can achieve the same accuracy as dense counterparts when initialized with an effective sparsity mask ("lottery ticket"). These findings emphasize the advantage of sparse weight configurations over dense matrices during training. While sparse training methods are theoretically more efficient, their practical application often results in lower accuracy compared to dense baselines. This discrepancy may be attributed to the challenges of identifying "lottery tickets" within a single training run. While sparse models reduce the FLOPs needed per step, we hypothesize that existing sparse training methods make sub-optimal use of these

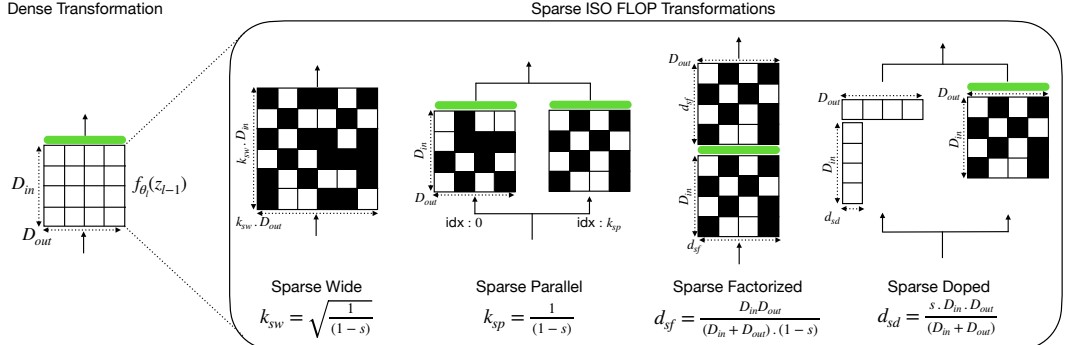

Figure 2: Different members of the Sparse-IFT family. Transformation of all members is parameterized by a single hyperparameter (i.e., sparsity level ($s$)). Black and white squares denote sparse and active weights, respectively. Green block indicates a non-linear activation function (e.g., BatchNorm, ReLU, LayerNorm). All transformations are derived with sparsity set to $50\%$ as an example, are Iso-FLOP to the dense feedforward function $f_{\theta_l}$, and hence can be used as a drop-in replacement of $f_{\theta_l}$. See Section 2.2 for more details about each member.

computational savings. For example, state-of-the-art sparse training methods [19, 37, 51, 83, 94] invest these FLOP savings into longer training schedules to close the accuracy gap and compensate for the inability to discover an optimal mask earlier in training. This setup is inefficient since it ultimately requires more training FLOPs than the dense baseline to reach the same target accuracy. In our work, we take an orthogonal approach and invest these FLOP savings into (a) increasing the representational capacity of a layer and (b) increasing its search space, which we hypothesize can facilitate the discovery of an optimal sparse mask [74, 80]. While utilizing larger sparsity-enabled models has exhibited accuracy improvement potential, the challenge lies in designing an appropriate architecture. For instance, when aiming to surpass the ResNet-18 performance on ImageNet, finding the right sparsity and larger network design is crucial. Many studies explore diverse combinations to balance sparsity and network size for outperforming dense models. However, these methods often lack FLOP efficiency, requiring multiple iterations for optimal settings and hyperparameter tuning. Therefore, we propose replacing dense transformations with FLOP-equivalent sparse transformations. We denote these transformations as the Sparse Iso-FLOP Transformation (Sparse-IFT) family.

## 2.1 Sparse Iso-FLOP Transformations

**Setup** For clarity, we will explain our method for a fully connected neural network. In Appendix A.1, we detail the straightforward extension of our method to convolutional layers. Let $\mathcal{N}$ denote a $L$ layered DNN parameterized by $\Theta_{\mathcal{N}}$. Let $\Theta_{\mathcal{N}} \in \{\theta_1, ..., \theta_L\}$ denote the parameters of the DNN. The output of the $l$-th layer is defined as: $z_l = \sigma(f_{\theta_l}(z_{l-1}))$ for some activation function $\sigma$ (e.g., ReLU [64]) and feedforward function $f_{\theta_l}$. Specifically, let $f_{\theta_l}(z_{l-1}) = \theta_l^T z_{l-1}$, where $\theta_l \in \mathbb{R}^{D_{in} \times D_{out}}$, $z_{l-1} \in \mathbb{R}^{D_{in} \times B}$ and $B$, $D_{in}$, $D_{out}$ denote the batch-size, input, and output dimensionality of features respectively. The total FLOPs needed for $f_{\theta_l}$ are given by $B \cdot D_{in} \cdot D_{out}$.

In the standard setup, the feedforward function $f_{\theta_l}$ computes the output features as a linear transformation of input features. From a theoretical perspective, the feedforward function can make use of arbitrary non-linear transformations. However, in practice, most transformations are expressed as dense matrix multiplications due to widespread support on GPUs [68]. As stated before, we are interested in improving the training efficiency of DNNs, by enhancing the representational capacity of the feedforward function. Naively increasing the representational capacity by stacking more layers [47], increasing width [95], mixture of experts [76], etc. increases the computational FLOPs. In our work, we use unstructured sparsity in weight matrices and ensure that the FLOPs of the transformation are the same as that of a dense feedforward function. Let $\Psi_l$ denote the set of Sparse Iso-FLOP Transformations (Sparse-IFT) for a particular layer $l$:

$$\Psi_l : \{\psi_l(s), 0 \le s < 1, g(\psi_l) \approx g(f_{\theta_l})\},$$

where $\psi_l$ is a transformation, $s$ represents the sparsity level, and $g(\cdot)$ returns the computational FLOPs. Each transformation in this set satisfies the following properties: (1) the computational FLOPs of the transformation $\psi_l$ are same as that of dense transformation $f_{\theta_l}$, and (2) the transformation is

parameterized by a single hyperparameter - the sparsity level. Since these transformations are Iso-FLOP to the dense feedforward function, we can use them as drop-in replacements without affecting the FLOPs of a layer. While there may be other FLOP-invariant transformations, in this work, we detail four different members: Sparse Wide, Sparse Parallel, Sparse Factorized, and Sparse Doped.

## 2.2 Members of Sparse-IFT

**Sparse Wide**  The sparse wide transformation augments the representational capacity of a layer by increasing the number of output features while keeping $s$ fraction of weights sparse. When using this transformation, we widen the input and output features for all the $L$ layers of the network with the same widening factor, $k_{sw}$, to avoid a mismatch in feature dimensionality across layers. Let $\theta_l^{sw} \in \mathbb{R}^{k_{sw} \cdot D_{in} \times k_{sw} \cdot D_{out}}$ denote the transformation matrix, with $s$ fraction of weights being sparse. Since the fraction of non-sparse weights is given by $1 - s$, the FLOPs required by this transformation are $B \cdot (k_{sw} \cdot D_{in}) \cdot (k_{sw} \cdot D_{out}) \cdot (1 - s)$. Setting these equal to the FLOPs of the original dense $f_{\theta_l}$, we obtain the widening factor $k_{sw} = \sqrt{\frac{1}{(1-s)}}$. If we set the sparsity $s$ to 0, we obtain $k_{sw}$ as 1 and recover the original dense feedforward function.

**Sparse Parallel**  The sparse parallel transformation replaces the feedforward function with a sum of $k_{sp}$ non-linear functions. Let $\theta_l^{sp} \in \{\theta_l^{sp,1}, ..., \theta_l^{sp,k_{sp}}\}$ denote the parameters of this transformation, where $\theta_l^{sp,j} \in \mathbb{R}^{D_{in} \times D_{out}}$ denotes the transformation matrix of $j^{th}$ function, where $s$ fraction of weights are sparse. The sparse parallel transformation in this case is $\psi_l^{sp} = \sum_{j=1}^{k_{sp}} \sigma((\theta_l^{sp,j})^T z_l)$, where $\sigma$ is a non linear function. In practice, $\psi_l^{sp}$ is implemented as a layer with $k_{sp}$ parallel branches. The computational FLOPs of this transformation is $k_{sp} \cdot B \cdot D_{in} \cdot D_{out} \cdot (1 - s)$. Setting these FLOPs equal to FLOPs of $f_\theta$, we obtain $k_{sp} = \frac{1}{(1-s)}$. Note, at $s = 0$, the number of parallel branches $k_{sp}$ is 1. If we replace the non-linear function $\sigma$ with Identity, we can recover the original dense feedforward transformation.

**Sparse Factorized**  The transformation matrix of the feedforward function $f_{\theta_l}$ is denoted by $\theta_l \in \mathbb{R}^{D_{in} \times D_{out}}$. Multiple works have explored matrix factorization techniques to express the transformation matrix $\theta_l$ as a product of two matrices $\theta_l = UV^T$, where $U \in \mathbb{R}^{D_{in} \times d}$, $V \in \mathbb{R}^{D_{out} \times d}$. Khodak et al. [39], Tai et al. [82] and Chen et al. [9] have explored low-rank factorization ($d << D_{out}$) as a form of structured sparsity to improve training and inference efficiency, while Arora et al. [1] and Guo et al. [26] have explored overparameterized factorizations for better generalization and faster convergence. In contrast, we use factorization to augment the representational capacity without decreasing or increasing the FLOPs. More precisely, let $\theta_l^{sf} \in \{U_l, V_l\}$ denote the parameters of this transformation, where $U_l \in \mathbb{R}^{D_{in} \times d_{sf}}$, $V_l \in \mathbb{R}^{d_{sf} \times D_{out}}$ are sparse matrices with $s$ fraction of their weights being sparse. The functional transformation in this case is $\psi_l^{sf} = V_l^T \sigma(U_l^T z_l)$. The computational FLOPs of this transformation is $d_{sf} \cdot B \cdot (D_{in} + D_{out}) \cdot (1 - s)$. Setting these FLOPs equal to FLOPs of $f_{\theta_l}$, we obtain $d_{sf} = \frac{D_{in} \cdot D_{out}}{(D_{in} + D_{out}) \cdot (1-s)}$. Note, setting sparsity $s = 0$, we recover a non-linear low-rank factorization with dense matrices.

**Sparse Doped**  family of transformation is inspired by works [3, 5, 85, 87] which approximate a dense matrix with a combination of low-rank factorization and sparse matrix. In our work, we replace the feedforward function with low-rank factorization (with rank $d_{sd}$) and an unstructured sparse weight matrix (with sparsity $s$). Let $U_l \in \mathbb{R}^{D_{in} \times d_{sd}}$, $V_l \in \mathbb{R}^{d_{sd} \times D_{out}}$ denote the low-rank matrices, and $\theta_l^{sd} \in \mathbb{R}^{D_{in} \times D_{out}}$ denote the matrix with unstructured sparsity. The functional transformation, in this case, is given by $\psi_l^{sd} = V_l^T (U_l^T z_l) + \sigma((\theta_l^{sd})^T z_l)$. The computational FLOPs associated with this transformation are $B \cdot d_{sd} \cdot (D_{in} + D_{out}) + (1 - s) \cdot B \cdot D_{in} \cdot D_{out}$. Setting these FLOPs equal to FLOPs of $f_{\theta_l}$, we obtain $d_{sd} = \frac{s \cdot D_{in} \cdot D_{out}}{(D_{in} + D_{out})}$. Note, as $s \to 0$ and $d_{sd} \to 0$, the low-rank component of the transformation disappears, and we can recover the dense feedforward function as a special case by setting $\sigma$ to Identity.

## 2.3 Cardinality of Search Space

One of our hypotheses is that increasing the search space of the sparsity mask via Sparse-IFT can make training more efficient. Results from past work support this hypothesis. Ramanujan et al. [74] demonstrate that the odds of finding a lottery ticket in a randomly initialized network

increase with the width of a network. Liu et al. [54] and Stosic and Stosic [80] show that increasing the search space by increasing width or depth improves accuracy. In our work, we define the cardinality of a search space as the number of weights a sparse training method can explore. Table 1 characterizes the cardinality of search space for each member of the Sparse-IFT family.

The search space for Sparse Wide, Sparse Parallel, and Sparse Factorized transformations increase proportional to the width scaling factor, number of parallel branches, and size of intermediate hidden dimension, respectively. Sparse Doped transformation splits its computational FLOPs between low-rank factorization and unstructured sparse weight matrix. The size of the unstructured weight matrix is invariant to sparsity; thus cardinality of search space for this transformation is constant.

Table 1: Cardinality of search space of sparsity mask for different members of the Sparse-IFT family.

| Transformation | Cardinality of Search Space |
|---|---|
| Sparse Wide | $(k_{sw})^2 \cdot (D_{in} \cdot D_{out})$ |
| Sparse Parallel | $k_{sp} \cdot (D_{in} \cdot D_{out})$ |
| Sparse Factorized | $d_{sf} \cdot (D_{in} + D_{out})$ |
| Sparse Doped | $D_{in} \cdot D_{out}$ |

## 3 Experiments

In this section, we demonstrate how transformations from the Sparse-IFT Family lead to improvements across a variety of different tasks in the CV and NLP domains. First, in Section 3.2, we describe the experimental setups and validate the design choices through multiple ablation studies on CIFAR-100 [40], followed by results on ImageNet [41]. Then, in Section 3.5, we highlight the advantages of pre-training with Sparse-IFT through gains on downstream tasks. Next, we present the benefits of Sparse-IFT in the NLP domain by demonstrating results on GPT [2] in Section 3.6. Finally in Section 4, we benchmark efficiency of Sparse-IFT using FLOPs, parameters and wall-clock time as metrics. Unless stated otherwise, the results presented below are obtained by replacing all dense layers with a given transformation from the Sparse-IFT family while only tuning the sparsity level. All sparse models are trained using a uniform sparsity distribution (i.e., all layers have the same sparsity level). We adopt the default hyperparameters from RigL [19] for dynamic sparsity. More details about the setup can be found in Appendix B.2.

### 3.1 Implementation Details

**Computer Vision**  We evaluate our method on CIFAR-100 and ImageNet using CNNs and hybrid Vision Transformer (ViT) networks. We follow published training settings for CIFAR-100 [15] and ImageNet [67]. For both datasets, we follow the standard evaluation procedures and report the top-1 accuracy. Details for model architectures, datasets, and training hyperparameters are given in Appendix B.2. All standard deviation was reported over 3 random seeds. For a few computationally expensive experiments, we report results from a single run due to budget constraints.

**Natural Language Processing**  We evaluate Sparse-IFT by training GPT-3 Small [2] from scratch on the WikiText-103 [60] language modeling task, a commonly used NLP benchmark dataset. The compute cost and resources for training quickly become prohibitive when transforming GPT models with Sparse-IFT. Hence, we train our GPT models on the Cerebras CS-2 [45, 46] and leverage its ability to accelerate training with unstructured sparsity. We provide more details about training-time performance in Section 4. Currently, Cerebras CS-2's specialized kernels support training with static unstructured sparsity; therefore, results in this section are reported without DST methods.

### 3.2 Results and Ablations on CIFAR-100

In this section, we conduct various ablations to validate our design choices. Unless stated otherwise, all experiments below are with ResNet-18 architecture on CIFAR-100.

**Importance of Dynamic Sparsity**  All members of the Sparse-IFT family utilize transformations with unstructured sparsity. This study investigates the importance of the sparse training method when training different configurations of Sparse-IFT architectures. For this analysis, we focus on the Sparse Wide IFT and evaluate it with transformations obtained with sparsity $\in \{50\%, 75\%, 90\%\}$ using three sparse training methods: static sparsity, SET [62] and RigL [19]. RigL and SET are dynamic sparse training methods in which the sparsity mask evolves during training. The key difference is that RigL updates the mask based on gradient information, whereas SET updates the

mask randomly. Results of our ablation are documented in Table 2. Here, the following trends can be observed: 1) the Sparse Wide IFT outperforms dense baselines across all operating points (sparsity and sparse training method), 2) dynamic sparse training methods (RigL and SET) obtain higher accuracies compared to training with static sparsity, and 3) gains with static sparsity plateau at lower levels of sparsity, while dynamic sparse training methods gain accuracy at higher sparsities.

As mentioned in Section 2.3, Sparse-IFT transformations increase the search space $\propto$ sparsity. Dynamic sparse training methods can explore and exploit this increased search space [80] and therefore outperform training with static sparsity. RigL consistently outperforms SET among the two dynamic sparse training methods we evaluated. Consequently, we adopt RigL as the sparse training method for all the experiments below.

Table 2: Sparse Wide IFT using various sparse training methods with ResNet-18 on CIFAR-100 across different levels of sparsity (columns). Best accuracy for each sparse training method is highlighted in bold.

| Dense | Sparse Method | 0.50 | 0.75 | 0.90 |
|---|---|---|---|---|
| | Static | **78.5 ± 0.3** | 78.3 ± 0.1 | 78.2 ± 0.3 |
| 77.0 ± 0.2 | SET | 78.8 ± 0.1 | 79.2 ± 0.2 | **79.8 ± 0.2** |
| | RigL | 79.1 ± 0.2 | 79.5 ± 0.1 | **80.1 ± 0.2** |

**Importance of Using Non-Linear Activations** Some members of the Sparse-IFT family are inspired by recent works which overparameterize the feedforward function during training and fold it back into a single dense matrix post training [16, 17, 18, 26]. Although these works show the benefits of linear overparameterization, this comes at the cost of a significant increase in training FLOPs. In contrast, while we also increase the representational capacity of the feedforward function, we do so with an Iso-FLOP transformation. Since we remain Iso-FLOP to the original dense model, we do not require post-training modifications to collapse weight matrices for inference efficiency. This uniquely allows us to use non-linearities (e.g., ReLU) in members of the Sparse-IFT family to enhance the representational capacity of the network further. We validate the importance of this design choice by training ResNet-18 with Sparse Factorized IFT with and without non-linearities, and observe significant accuracy gains across all sparsity levels when using non-linear activations. For example, at 90% Sparse Factorized, using non-linearity, we see a 1.8% gain in test accuracy over the ResNet-18 CIFAR-100 dense baseline, compared to a drop of 0.5% without it. These findings hold for other members of the Sparse-IFT family as well (see Appendix B.1 for more details).

**Sparse-IFT ResNet-18** Here, we evaluate different members of the Sparse-IFT family on ResNet-18 and CIFAR-100 across different sparsity levels. Table 3 highlights the best accuracy achieved by each member of the Sparse-IFT family. Compared to the accuracy of the dense baseline (77%),

Table 3: Sparse-IFT families on CIFAR-100 with ResNet-18 model across different levels of sparsity (columns). Best accuracy of each transformation is highlighted in bold.

| Dense | Transformation | 0.50 | 0.75 | 0.90 |
|---|---|---|---|---|
| | Sparse Wide | 79.1 ± 0.2 | 79.5 ± 0.1 | **80.1 ± 0.2** |
| 77.0 ± 0.2 | Sparse Factorized | 77.8 ± 0.2 | 78.4 ± 0.5 | **78.9 ± 0.5** |
| | Sparse Parallel | 77.9 ± 0.4 | **79.1 ± 0.2** | 78.2 ± 0.2 |
| | Sparse Doped | **78.2 ± 0.1** | 77.8 ± 0.1 | 76.9 ± 0.2 |

all Sparse-IFT members obtain significant accuracy improvements using the same FLOPs as the dense model. We note that the Sparse Doped transformation is the only member of the Sparse-IFT family which does not gain accuracy at higher levels of sparsity. We hypothesize that this phenomenon occurs due to two reasons: (a) cardinality of the search space of the sparsity mask does not increase with sparsity level (see Table 1), and (b) the number of active weights in the unstructured matrix decreases $\propto$ sparsity. In Appendix B.3.1, we compare Sparse-IFT against other baselines obtained with sparse training methods (e.g., RigL and SET) under the same training efficiency setup. Specifically, we train ResNet-18 model on CIFAR-100 at sparsity levels $\in \{50\%, 75\%, 90\%\}$, and ensure that these runs use the same FLOPs as the dense baseline by extending the training iterations. Our results show that Sparse-IFT outperforms these competitive baselines by a significant margin.

**Sparse-IFT vs. Dense Overparametrization** The success of Sparse-IFT members can be attributed to efficient exploration of large search space with sparsity. Training this large search space in a dense manner leads to consumption of more training FLOPs than the dense baseline, but provides

us with the upperbound (in terms of accuracy) for a sparse subnetwork. In this section, we will characetrize this gap between the Sparse-IFT members and their dense counterpart. In Table 4, we compare the sparse and dense counterparts of the two best performing Sparse-IFT members.

Both dense and sparse training yield similar accuracy across all sparsity levels, demonstrating efficient exploration and exploitation of over-parameterized space without the computational cost of dense training. For instance, dense runs (using Sparse-IFTs at 90% sparsity) require 10x more FLOPs than sparse runs.

Table 4: Sparse-IFTs trained in a sparse and dense manner on CIFAR-100 with ResNet-18 for different levels of sparsity.

| Transformation | Train Method | 0.50 | 0.75 | 0.90 |
|---|---|---|---|---|
| Sparse Wide | Sparse | $79.1 \pm 0.2$ | $79.5 \pm 0.1$ | $\mathbf{80.1 \pm 0.2}$ |
| | Dense | $78.9 \pm 0.2$ | $79.7 \pm 0.1$ | $\mathbf{80.2 \pm 0.3}$ |
| Sparse Parallel | Sparse | $77.9 \pm 0.4$ | $\mathbf{79.1 \pm 0.2}$ | $78.2 \pm 0.2$ |
| | Dense | $78.1 \pm 0.2$ | $\mathbf{78.9 \pm 0.1}$ | $78.1 \pm 0.1$ |

**Unstructured vs. Structured Sparsity**
We compare unstructured sparsity to structured sparsity with Sparse-IFT. In theory, for a fixed number of non-zero elements in a sparse mask, the use of unstructured sparsity can search over all the possible variations of the mask. However, since most hardware accelerators are not able to accelerate computations with unstructured sparsity, multiple

Table 5: Sparse Wide IFT with unstructured and structured sparsity across different levels of sparsity (columns) on CIFAR-100 with ResNet-18.

| Dense | Sparsity Pattern | 0.50 | 0.75 | 0.90 |
|---|---|---|---|---|
| $77.0 \pm 0.2$ | Unstructured | 79.1 | 79.5 | **80.1** |
| | N:M Block Sparse | 77.1 | **78.4** | 78.1 |

works have investigated training with structured sparsity (e.g., low-rank and block-sparse matrices) to obtain wall-clock speed-ups [6, 9, 13, 34, 39, 82]. We study structured sparsity by deriving Iso-FLOP configurations using low-rank and block sparsity with Sparse Wide IFT. We use the method proposed in Hubara et al. [34] to search N:M transposable sparsity, which can accelerate training on GPUs with Tensor Cores. In our evaluation, the low-rank factorization results were worse than block sparsity (see more details in Appendix B.3.3). Table 5 compares unstructured sparsity to block sparsity. Although using Sparse-IFT with block sparse matrices lead to improvements over the dense baseline, unstructured sparsity achieves the highest gains. This result can be explained by the fact that block-sparse matrices have reduced mask diversity [34] compared to unstructured sparse matrices.

### 3.3 Results with Efficient Architectures

To further understand the robustness of Sparse-IFT across different model families, we evaluate Sparse-IFT on architectures that are optimized for efficient inference (MobileNetV2 [75] and MobileViT [59]) and efficient training (BotNet [79]). We transform the dense layers in these architectures with Sparse Wide IFT and evaluate them at different sparsity levels. We observe a noticeable increase in test accuracy across

Table 6: Sparse Wide IFT with various efficient architectures on CIFAR-100 across different levels of sparsity (columns).

| | Dense | 0.50 | 0.75 |
|---|---|---|---|
| MobileNetV2 | $72.4 \pm 0.2$ | 73.4 | **73.7** |
| MobileViT-S | $73.5 \pm 0.1$ | 74.6 | **74.8** |
| BotNet-50 | $79.8 \pm 0.2$ | 80.3 | **80.6** |

all architectures (see Table 6). In addition, we demonstrate the robustness of the Sparse-IFT family by also applying the Sparse Parallel transformation and show consistent improvement across all architectures (see Appendix B.3.2). We evaluate the best performing architecture (BotNet-50) on ImageNet (see Section 3.4). The details of the experimental setup can be found in Appendix B.2.

### 3.4 Results on ImageNet

We take the best performing Sparse-IFT transformations (i.e., Sparse Wide IFT and Sparse Parallel IFT) on CIFAR-100, and evaluate them on ImageNet using ResNet-18. Both families of Sparse-IFT obtain significantly higher accuracy compared to the

Table 7: Sparse-IFT on ImageNet. Best result for each transformation and architecture is highlighted in bold.

| Model | Dense | Transformation | Sparsity | | |
|---|---|---|---|---|---|
| | | | 0.50 | 0.75 | 0.90 |
| ResNet-18 | $70.9 \pm 0.1$ | Sparse Wide | 72.7 | 73.8 | **74.4** |
| | | Sparse Parallel | 72.7 | 73.2 | **74.0** |
| ResNet-34 | $74.2 \pm 0.1$ | Sparse Wide | 75.6 | 76.4 | **76.8** |
| BotNet-50 | $77.5 \pm 0.1$ | Sparse Wide | 77.9 | 78.3 | **78.5** |

dense baseline (refer to Table 7). Note, Sparse Wide IFT ResNet-18 at 90% sparsity improves over the dense baseline by 3.5%, and is able to match accuracy of dense ResNet-34 with $2\times$ fewer training FLOPs (see Figure 1). We take the best performing transformation (Sparse Wide IFT) and apply it to ResNet-34 and BotNet-50. Increasing sparsity leads to a consistent increase in accuracy, indicating improved training efficiency at higher sparsities. On BotNet-50, a hybrid ViT model, we see a 1% improvement at 90% sparsity.

### 3.5 Transfer Learning with Sparse-IFT

To show the effectiveness of pre-training our Sparse-IFT classification backbones, we evaluate them on 1) object detection on MS COCO 2017 [48], and 2) semantic segmentation on CityScapes [12]. For object detection, we adopt the RetinaNet [50] framework from the MMDetection open-source toolbox [7] and report results in the standardized training setting. For semantic segmentation, we utilize DeepLabV3+ [8] in the MMSegmenation open-source toolbox [11]. We evaluate

Table 8: Sparse-IFT variants of ResNet-18 as backbones for : (a) Object detection on MS COCO, (b) Semantic segmentation on Cityscapes.

| | Metric | Dense | Sparsity | | |
| | | | 0.50 | 0.75 | 0.90 |
|---|---|---|---|---|---|
| MS COCO | AP | 29.3 | 31.3 | 32.8 | **34.5** |
| | $AP_{50}$ | 46.2 | 49.0 | 51.0 | **53.5** |
| | $AP_{75}$ | 30.9 | 33.0 | 34.8 | **36.5** |
| CityScapes | mIoU | 76.7 | 77.9 | 78.9 | **79.1** |
| | mAcc | 84.4 | 85.1 | 85.7 | **86.0** |

ResNet-18 with Sparse Wide IFT (best-performing transformation on ImageNet). To ensure FLOP-equivalent comparisons with the dense backbone, the Sparse-IFT backbones remain sparse during fine-tuning. Appendix B.3.4 provides more details regarding the training setup. We summarize our findings in Table 8, where using Sparse Wide IFT ResNet-18 backbone leads to significant accuracy gains across all metrics on both downstream tasks.

### 3.6 Results on GPT End-to-End Training

We train the Sparse Wide IFT GPT-3 Small models at 50% and 75% sparsity levels, and compare against the standard dense GPT-3 Small and GPT-3 Medium models. Following Dao et al. [13], we train all models from scratch on the WikiText-103 dataset and report the average test perplexity (PPL) over 3 random seeds

Table 9: Sparse-IFT for pre-training GPT-3 Small from scratch on WikiText-103 and report the test perplexity (lower is better).

| | Dense | 0.50 | 0.75 |
|---|---|---|---|
| GPT-3 Small | $20.8 \pm 0.3$ | **20.4** | 22.1 |

in Table 9. We show that Sparse Wide IFT GPT-3 Small at 50% sparsity improves the perplexity by 0.4 over its dense counterpart. We also note that the Sparse Wide IFT GPT-3 Small model performs comparable to a dense GPT-3 Medium ($20.5 \pm 0.2$ PPL) while using 2.4x fewer training FLOPs. In Appendix C.1, we provide details on the hyperparameters and how the total training FLOPs for the models in Table 9 were calculated.

**GPT Pre-training and Fine-tuning**   While not our main focus, it is worth noting that Sparse-IFT can be used for fine-tuning NLP models. After sparse pre-training, the Sparse-IFT model can undergo fine-tuning while remaining sparse or after densifying through techniques like SPDF [86]. Preliminary fine-tuning experiments on BERT and GPT, with detailed results in Appendixx C.2.

## 4 Benchmarking Efficiency of Sparse-IFT

Model training efficiency can be characterized by FLOPs, parameters, or wall-clock time. While wall-clock time is an ideal metric for benchmarking model configurations, it can be influenced by external factors like hardware design, memory bandwidth, computational capabilities, kernel support, and operation types. Hence, many studies opt to compare models using FLOPs or parameters instead. In our work, we chose FLOPs as the metric for two reasons: (a) to be comparable to existing sparsity work [19, 37, 58, 62], and (b) FLOPs correlate better with run-time compared to number of parameters. Through various studies in Section 3, we have demonstrated improved training efficiency of Sparse-IFT w.r.t training FLOPs. In Appendix D, we benchmark the efficiency of Sparse-IFT in relation to model parameters and time. Our results show that Sparse-IFT variants perform well in inference, offering higher accuracy for a fixed parameter budget. When supported by platforms with

unstructured sparsity acceleration, Sparse-IFT also provides speed advantages during both training and inference.

## 5    Related Work

Our work is similar to the body of work studying the role of overparameterization and sparsity for training DNNs. The modeling capacity needed to learn a task is often unknown. Hence, we often solve this by training overparameterized models to fully exploit the learning capability and then compress them into a smaller subnetwork.

**Overparameterization**    Nakkiran et al. [65] show that DNNs benefit from overparameterization. Following this, there have been many works that leverage overparameterization by scaling the size of models [25, 71] and augmenting existing DNNs to increase modeling capacity and the accuracy of trained networks  [4, 16, 18, 27, 53, 88]. These methods use linear parameterizations of the model, making them highly inefficient to train, and are focused on improving inference throughput (reduced latency). In contrast, our work is focused on improving the modeling capacity using sparse non-linear parameterizations. Our approach enhances accuracy without increasing training FLOPs compared to the baseline model, and while still maintaining equivalent inference FLOPs.

**Sparse Training**    The Lottery Ticket Hypothesis [20, 21] shows that accurate sparse subnetworks exist in overparameterized dense networks but require training a dense baseline to find.  Other approaches have proposed frameworks for identifying lottery tickets [58, 98] but still require a lot of compute resources. Following this, various attempts have been made to find the optimal sparse subnetwork in a single shot. These methods either try to find the subnetworks at initialization [14, 44, 84, 90] or dynamically during training [19, 37, 62, 72]. However, given a fixed model capacity, these methods tradeoff accuracy relative to the dense baseline to save training FLOPs.  Stosic and Stosic [80] and Ramanujan et al. [74] increase the search space during sparse training to retain accuracy; however, do not guarantee FLOPs savings. In contrast to these methods, our work introduces a set of non-linear sparse transformations, which increase the representational capacity of the network. This approach does not introduce a new sparse training algorithm, but instead improves the search space of existing methods, leading to improved generalization while being efficient to train.

**Iso-Parameter vs. Iso-FLOP**    Recent sparsity literature is focused on improving generalization at high sparsity levels. Hence, layer-wise sparsity distributions such as the Erdös-Rényi-Kernel [19], Ideal Gas Quota [10], and parameter leveling [24] are often used with sparse training to boost accuracies. However, these works target the setting where the models being compared have a fixed parameter budget (i.e., Iso-Parameter), which does not translate to similar training FLOPs to the original dense model (especially in CNNs). As a result, training models with these distributions often require different memory or computational resources per layer. Our approach does not focus on this Iso-Parameter setting but instead adopts the uniform sparsity distribution (i.e., every layer gets the same sparsity level), ensuring uniform FLOP reductions across the network. We achieve equivalent computational FLOPs to a dense network through our Iso-FLOP transformations and sparsity.

## 6    Conclusion

We introduce a new family of Sparse Iso-FLOP Transformations (Sparse-IFT) to improve the training efficiency of DNNs. These transformations can be used as drop-in replacements for dense layers and increase the representational capacity while using sparsity to maintain training FLOPs. This increase in capacity also translates to a larger search space allowing sparse training methods to explore better and identify optimal sparse subnetworks. For the same computational cost as the original dense model, Sparse-IFT improves the training efficiency across multiple model families in the CV and NLP domains for various tasks. A limitation of our work is that most of the current hardware accelerators do not support unstructured sparsity. We hope our results along with the promising benchmarks (Section 4) on the Cerebras CS-2 and Neural Magic DeepSparse runtime will motivate the industry to build better support for unstructured weight sparsity during training and inference.

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

# A  Additional Methodology Details

## A.1  Sparse-IFT for Convolutional Layers

In this section, we detail the straightforward extension of the Sparse-IFT family for convolutional layers.

**Sparse Wide**    Similar to the setup for fully connected layers, in the case of convolutional layers, we widen the number of input and output channels.

**Sparse Parallel**    Similar to the setup for fully connected layers, in the case of convolutional layers, we can implement this transformation with the use of convolutional branches in parallel.

**Sparse Factorized and Sparse Doped**    Let $\theta_l \in \mathbb{R}^{c_{in} \times c_{out} \times k_h \times k_w}$ represent the weight matrix of a convolutional layer, where $c_{in}, c_{out}, k_h, k_w$ denote the input channels, output channels, kernel height, and kernel width, respectively. We apply low-rank or matrix factorization to the weight matrix by first converting the 4D tensor into a 2D matrix with shape: $(c_{in} \cdot k_h \cdot k_w) \times c_{out}$. In this setup, we can express $\theta_l = UV^T$, where $U \in \mathbb{R}^{c_{in} \cdot k_h \cdot k_w \times d}$, $V \in \mathbb{R}^{c_{out} \times d}$. In this factorization, $U$ learns a lower-dimensional set of features and is implemented as a convolutional layer with $d$ output channels and $k_h \times k_w$ filter. $V$ matrix expands this low-dimensional set of features and is implemented as a convolutional layer with $1 \times 1$ filter.

### A.1.1  Sparse-IFT for Depthwise Convolution Layers

For a normal convolution layer, all inputs are convolved to all outputs. However, for depthwise convolutions, each input channel is convolved with its own set of filters. Let $\theta_l \in \mathbb{R}^{c_{in} \times c_{out} \times k_h \times k_w}$ represent the weight matrix of a normal convolution layer, where $c_{in}, c_{out}, k_h, k_w$ denote the input channels, output channels, kernel height, and kernel width, respectively. An equivalent depthwise convolution layer will have weights $\theta_{dw,l} \in \mathbb{R}^{1 \times c_{out} \times k_h \times k_w}$.

**Sparse Wide**    A Sparse Wide depthwise convolution will have weights $\theta_{dw,l}^{sw} \in \mathbb{R}^{1 \times k_{sw} \cdot c_{out} \times k_h \times k_w}$. Since the fraction of non-sparse weights is given by $1 - s$, the FLOPs required by this transformation are $B \cdot (k_{sw} \cdot c_{out}) \cdot k_h \cdot k_w \cdot (1 - s)$. Setting these equal to the FLOPs of the original dense $\theta_{dw,l}$, we obtain the widening factor $k_{sw} = \frac{1}{(1-s)}$. In this case, we do not scale the input channels as it converts the depthwise convolution to a grouped convolution without an equivalent scaling in the number of groups.

**Other Sparse-IFT Transformations**    The Sparse Wide IFT generally changes a layer's input and output channels, subsequently scaling the following layers in a CNN. However, the other Sparse-IFT transforms (Sparse Parallel, Sparse Factorized, and Sparse Doped) do not modify a convolution layer's input or output channels (as seen in Figure 2). This allows for fine-grained control of what layers to apply the Sparse-IFT transformations. Since depthwise convolutions are an extreme form of structured sparsity, where some filters interact with only specific input channels, we opt not to sparsify them when using the other Sparse-IFT transformations and leave the layer unchanged while still maintaining FLOPs equivalent to the dense baseline. Note that the different convolution layers surrounding the depthwise convolution are still transformed with Sparse-IFT to increase their representational capacity.

# B  Computer Vision: Experimental Settings

## B.1  Importance of Non-linearity

We use BatchNorm [35] followed by ReLU [64] as a non-linearity. We provide an extended set of empirical results in Table 10 to help validate the importance of training with and without non-linearity by training configurations of the Sparse Parallel, Factorized, and Doped IFT families at different levels of sparsity. The results without non-linear activation functions are often worse than the dense accuracy (77%) across all Sparse-IFT family transformations. We omit Sparse Wide in Table 10

because here we increase the number of channels in the convolutional layers while maintaining the existing architecture.

Table 10: Evaluation on the importance of utilizing the non-linear activation across different members of Sparse-IFT with ResNet-18 on CIFAR100 across different values of sparsity (columns). Non-linear activations enhance the representational capacity of Sparse-IFT, leading to higher accuracy. All reported results are the average over 3 random seeds.

| Transformation | Non-linear activation | 0.50 | 0.75 | 0.90 |
|---|---|---|---|---|
| Sparse Factorized | ✗ | $75.9 \pm 0.3$ | $76.6 \pm 0.4$ | $76.5 \pm 0.4$ |
|  | ✓ | $\mathbf{77.8 \pm 0.4}$ | $\mathbf{78.4 \pm 0.5}$ | $\mathbf{78.9 \pm 0.5}$ |
| Sparse Parallel | ✗ | $77.1 \pm 0.1$ | $77.2 \pm 0.2$ | $77.6 \pm 0.1$ |
|  | ✓ | $\mathbf{77.9 \pm 0.2}$ | $\mathbf{79.1 \pm 0.2}$ | $\mathbf{78.2 \pm 0.2}$ |
| Sparse Doped | ✗ | $77.3 \pm 0.2$ | $77.1 \pm 0.1$ | $76.5 \pm 0.2$ |
|  | ✓ | $\mathbf{78.2 \pm 0.1}$ | $\mathbf{77.8 \pm 0.1}$ | $\mathbf{76.9 \pm 0.2}$ |

## B.2 Computer Vision: Pre-Training Settings

**CIFAR-100**   Our implementation of CIFAR-100 follows the setup from [15] for ResNets. We train the models for 200 epochs with batches of 128 using SGD, Nesterov momentum of 0.9, and weight-decay of $5 \times 10^{-4}$. The learning rate is initially set to 0.1 and is scheduled to decay to decrease by a factor of 5x after each of the 60th, 120th, and 160th epochs. Following recent advances in improving ResNets, we initialize the network with Kaiming He initialization [30], zero-init residuals [31], and disable weight-decay in biases and BatchNorm [35] layers. For CIFAR-100 experiments with MobileNetV2, MobileViT-S, and BotNet-50, we follow the same training setup used for ResNet, but the learning rate is scheduled via cosine annealing.

**ImageNet**   Our implementation of ImageNet follows the standard setup from [42, 78]. The image is resized with its shorter side randomly sampled in [256, 480] for scale augmentation [78]. A 224 $\times$ 224 crop is randomly sampled from an image or its horizontal flop, and then normalized. For evaluation, the image is first resized to 256 $\times$ 256, followed by a 224 $\times$ 224 center crop, and then normalized. Following recent advances in improving ResNets, we initialize the network with Kaiming He initialization [30] and zero-init residuals [31].

For ResNets, we replicate the settings recommended by Nvidia [67], which uses the SGD optimizer with a momentum of 0.875 and weight decay of $3.0517578125 \times 10^{-5}$. We disable weight-decay for biases and BatchNorm layers. The model is trained with label smoothing [81] of 0.1 and mixed precision [61] for the standard 90 epochs using a cosine-decay learning rate schedule with an initial learning rate of 0.256 for a batch size of 256. Srinivas et al. [79] follow the same setup as ResNet for training BotNet-50 on ImageNet, therefore we maintain the same hyperparameter settings as Nvidia [67] for our BotNet-50 ImageNet experiments.

**Sparsity Setup**   For enabling the Sparse-IFT transformations, we use the RigL [19] algorithm in its default hyperparameter settings ($\alpha = 0.3, \Delta T = 100$), with the drop-fraction ($\alpha$) annealed using a cosine decay schedule for 75% of the training run. We keep the first and last layers (input convolution and output linear layer) dense to prevent a significant degradation in model quality during pre-training, which is standard practice. We account for these additional dense FLOPs by increasing the sparsity in the remaining layers, similar to Gale et al. [22] and Liu et al. [54].

## B.3 Computer Vision

### B.3.1 Sparse-IFT vs. Extended Sparse Training Schedules

We provide a direct comparison with sparse training methods (e.g., RigL and SET) in the Iso-FLOP setting (i.e., training with a longer schedule) to demonstrate the significance of our results with respect to this standard sparse baselines. As shown in the Table 11, Sparse-IFTs outperform dynamic sparse training methods by a significant margin across all levels of sparsity. Note, at higher levels of sparsity (e.g., 90%), sparse training methods obtain worse accuracy compared to the FLOP equivalent

Table 11: Results with ResNet-18 on CIFAR-100 across different values of sparsity (columns). Best accuracy for each sparse training method is highlighted in bold. The original dense ResNet-18 model obtains an accuracy of 77.0±0.2. All reported results are over 3 random seeds.

| Dense | Transformation | Sparse Training Method | Epochs | 0.50 | 0.75 | 0.90 |
|---|---|---|---|---|---|---|
| | Sparse Wide | SET | $200 \cdot \frac{1}{1-s}$ | **78.7 ± 0.2** | 78.4 ± 0.1 | 76.8 ± 0.1 |
| 77.0 ± 0.2 | Sparse Wide | RigL | $200 \cdot \frac{1}{1-s}$ | **78.9 ± 0.1** | 78.8 ± 0.1 | 76.4 ± 0.2 |
| | Sparse Parallel | RigL | 200 | 79.1 ± 0.2 | 79.5 ± 0.1 | **80.1 ± 0.2** |

dense baseline. In contrast, with Sparse-IFT, we observe higher accuracy across all levels of sparsity evaluated.

### B.3.2 Sparse-IFT on Efficient Computer Vision Architectures

Here, we provide an extended set of results on MobileNetV2, MobileViT-S, and BotNet-50 on CIFAR-100. In particular, we enable Sparse Wide and Sparse Parallel IFT at 50% and 75% sparsity values (see Table 12).

Table 12: Evaluation of Sparse Wide and Sparse Parallel IFT with various compute efficient architectures on CIFAR-100 across different values of sparsity (columns). Using Sparse Parallel IFT, all architectures outperform the dense baseline by a significant margin.

| | Dense | Transformation | 0.50 | 0.75 |
|---|---|---|---|---|
| MobileNetV2 | 72.4 ± 0.2 | Sparse Wide | 73.4 | **73.7** |
| | | Sparse Parallel | 72.9 | **73.3** |
| MobileViT-S | 73.5 ± 0.1 | Sparse Wide | 74.6 | **74.8** |
| | | Sparse Parallel | 73.7 | **74.4** |
| BotNet-50 | 79.8 ± 0.2 | Sparse Wide | 80.3 | **80.6** |
| | | Sparse Parallel | 79.7 | **80.5** |

### B.3.3 Evaluation of Sparse-IFT with Structured Sparsity

**Block Sparsity**   To derive Iso-FLOP configurations with block sparsity, we reuse the analysis done previously with unstructured sparsity (see Section 2.2) and express the width scaling as a function of sparsity. However, we will search for a block sparse mask during training instead of an unstructured sparsity mask. We use the method proposed by Hubara et al. [34] to search N:M transposable sparsity, which can accelerate both the forward and backward pass during training on NVIDIA GPUs with Tensor Cores. We use 4:8-T, 2:8-T, and 1:8-T block patterns to obtain 50%, 75%, and 87.5% sparsity, respectively. Note the 1:8-T block is the closest approximation to a 90% sparsity pattern attainable with a block size of 8. We also set up and experimented using the method proposed by Jiang et al. [38] to train with fine-grained sparse block structures dynamically. However, the algorithm uses agglomerative clustering which led to a much slower runtime and quickly ran out of memory even at 50% sparsity using the Sparse Wide IFT on a single Nvidia V100 (16 GB).

**Low Rank**   Let $k_{lr}$ be the factor with which we widen all layers' input and output dimensions for low-rank factorization. We replace all dense layers with low-rank factorization, i.e. $\theta_l^{lr} = U_l V_l^T$, where $U_l \in \mathbb{R}^{(k_{lr} \cdot D_{in}) \times d}$ and $V_l \in \mathbb{R}^{(k_{lr} \cdot D_{out}) \times d}$. Given a widening factor and equating the FLOPs of this transformation to that of a dense transformation $f_\theta$, we obtain the following expression for rank $d$: $\frac{D_{in} \cdot D_{out} \cdot k_{lr}}{(D_{in} + D_{out})}$. We evaluate this factorization across different values of width-scaling $k_{lr}$ in Table 13.

### B.3.4 Evaluation on downstream tasks

#### COCO Object Detection

This dataset contains 118K training, 5K validation (`minival`), and 20K test-dev images. We adopt the standard single-scale training setting [49] where there is no additional data augmentation beyond

Table 13: Comparison of structured sparse and unstructured sparse methods on CIFAR-100 test accuracy on ResNet-18.

| Transformation | Sparsity Type | Sparsity | Width Scaling Factor | | | |
|---|---|---|---|---|---|---|
| | | | 1x | 1.41x | 2x | 3.16x |
| Low Rank, Linear | Structured | 0% | 74.1 | 74.3 | 74.3 | 73.4 |
| Low Rank, Non-Linear | Structured | 0% | 76.8 | 76.5 | 76.0 | 75.3 |
| Sparse Wide | N:M Block Sparse [34] | 4:8-T | | 77.1 | | |
| | | 2:8-T | | | **78.4** | |
| | | 1:8-T | | | | 78.1 |
| | Unstructured Sparse [19] | 50% | | 79.1 | | |
| | | 75% | | | 79.5 | |
| | | 90% | | | | **80.1** |

standard horizontal flipping. For training and testing, the input images are resized so that the shorter edge is 800 pixels [49]. The model is trained with a batch size of 16, using the SGD optimizer with a momentum of 0.9 and weight decay of $1 \times 10^{-4}$. We follow the standard 1x schedule (12 epochs) using a step learning rate schedule, with a 10x decrease at epochs 8 and 11, an initial learning rate warmup of 500 steps starting from a learning rate of $2 \times 10^{-5}$, and a peak learning rate of 0.01.

Table 14: Object detection results on COCO `minival` in the RetinaNet framework. Sparse Wide IFT configurations of RetinaNet outperform the dense baseline by a large margin on all metrics while using similar FLOPs.

| Backbone | AP | $AP_{50}$ | $AP_{75}$ | $AP_S$ | $AP_M$ | $AP_L$ |
|---|---|---|---|---|---|---|
| Dense | 29.3 | 46.2 | 30.9 | 14.7 | 31.5 | 39.6 |
| Sparse Wide (50%) | 31.3 | 49.0 | 33.0 | 16.6 | 34.0 | 42.0 |
| Sparse Wide (75%) | 32.8 | 51.0 | 34.8 | 17.3 | 35.8 | 43.3 |
| Sparse Wide (90%) | **34.5** | **53.5** | **36.5** | **18.6** | **37.6** | **45.3** |

### CityScapes Semantic Segmenation

**Setup** We follow the same training protocol as [97], where the data is augmented by random cropping (from $1024 \times 2048$ to $512 \times 1024$), random scaling in the range [0.5, 2], and random horizontal flipping. The model is trained with a batch size of 16, using the SGD optimizer with a momentum of 0.9 and weight decay of $5 \times 10^{-4}$. We follow the 80K iterations setup from MMSegmentation with an initial learning rate of 0.01 annealed using a poly learning rate schedule to a minimum of $1 \times 10^{-4}$. Similar to most setups that tune hyperparameters [55, 91, 97] for reporting the best results, we tune the learning rate for all our models. All our results are reported using a learning rate of 0.03 for the sparse backbones and 0.01 for the dense baseline.

Table 15: Semantic segmentation results on the Cityscapes `val` set using DeepLabV3+. Sparse Wide IFT configurations ResNet-18 backbones outperform the dense baseline on all metrics while using similar FLOPs.

| Backbone | mIoU | mAcc |
|---|---|---|
| Dense | 76.72 | 84.40 |
| Sparse Wide (50%) | 77.90 | 85.12 |
| Sparse Wide (75%) | 78.92 | 85.68 |
| Sparse Wide (90%) | **79.10** | **86.01** |

## C  Natural Language Processing: Experimental Settings

### C.1  Details for GPT End-to-End Training

Our end-to-end training setup for GPT-3 on WikiText-103 follows a similar procedure to Dao et al. [13]. We use a batch size of 512 and train with the AdamW optimizer for 100 epochs. Also, we use a

learning rate warmup for 10 epochs and a weight decay of 0.1. To discover good hyperparameters, we perform a grid search to discover an appropriate learning rate among {8e-3, 6e-3, 5.4e-3, 1.8e-3, 6e-4, 2e-4, 6e-5} that led to the best perplexity for a given compute budget on the validation set. In Table 16, we outline the architecture configurations for the original dense model and its Sparse Wide IFT 50% and 75% variants.

Table 16: Sizes and architecture definitions of the dense GPT-3 Small model and its Sparse Wide IFT variants.

| Model | Transformation | Sparsity | $n_{layers}$ | $d_{model}$ | $d_{\text{ff}}$ | $n_{heads}$ | $d_{head}$ |
|-------|---------------|----------|----------|----------|--------|----------|--------|
| GPT-3 Small | Dense | 0% | 12 | 768 | 3072 | 12 | 64 |
| GPT-3 Small | Sparse Wide | 50% | 12 | 1092 | 4344 | 12 | 64 |
| GPT-3 Small | Sparse Wide | 75% | 12 | 1536 | 6144 | 12 | 64 |

**WikiText-103 End-to-End Training Results**   We highlight that in Table 17, the Sparse Wide IFT GPT-3 Small at 50% sparsity attains a better perplexity on WikiText-103 while using 2.4x fewer training FLOPs than the GPT-3 Medium dense model. In this setup, using Sparse Wide transformation does not change the FLOP of the dense layer, but this leads to a slight increase in the attention FLOPs. This explains the 1.17x increase in FLOPs between the GPT-3 Small Sparse Wide at 50% sparsity and the dense GPT-3 Small model. Note, out of all the Sparse-IFT transformations, this increase only occurs in the Sparse Wide IFT.

Table 17: Details on the total training FLOPs for each GPT-3 model tested. We note that the reported FLOPs per sequence (seq) include both forward and backward passes. The reported perplexity (lower is better) is on the WikiText-103 test set over 3 random seeds.

| Model | Transformation | Sparsity | Total Seqs | Total FLOPs/ Seq | Total FLOPs | Total exaFLOPs | Perplexity |
|-------|---------------|----------|-----------|------------------|-------------|----------------|-----------|
| GPT-3 Small | Dense | 0% | 2.28e6 | 8.763e11 | 2.0011e18 | 2.00 | 20.8 ± 0.3 |
| GPT-3 Small | Sparse Wide | 50% | 2.28e6 | 1.029e12 | 2.3498e18 | 2.35 | **20.4 ± 0.2** |
| GPT-3 Medium | Dense | 0% | 2.28e6 | 2.4845e12 | 5.6734e18 | 5.67 | 20.5 ± 0.2 |

## C.2  Details for Sparse Pre-training and Dense Fine-tuning [86]

We provide an extended set of results that showcase the added benefit of using Sparse-IFT transformations. Here, we apply the Sparse Pre-training and Dense Fine-tuning (SPDF) framework introduced by Thangarasa et al. [86]. In this setup, all models are pre-trained under a similar FLOP budget. However, during the fine-tuning stage, Sparse-IFT models have extra representational capacity which can be enabled by allowing the zeroed weights to learn (i.e., dense fine-tuning). Even though the fine-tuning FLOPs are more than the original dense model, we leverage Sparse-IFT method's extra capacity to obtain accuracy gains on the downstream task. To ensure a fair baseline, we also compare dense fine-tuning to sparse fine-tuning (i.e., pre-trained model remains as-is) similar to Thangarasa et al. [86].

### C.2.1  SPDF on BERT

**Experimental Setup**   We train BERT models using the open-source LAMB [93] implementation provided by Nvidia [66]. In this setup, BERT is pre-trained on the BookCorpus [99] and Wikipedia datasets in two phases. In the first phase, models are trained for 82% of total iterations with a sequence length of 128. In the second phase, models are trained for the remaining 18% of iterations with sequence length 512. We use a batch size of 8192 and 4096 in phase 1 and phase 2, respectively. Table 18 shows details of the size and architecture of the BERT Small model. For finetuning models on SQuADv1.1 [73], we train for two epochs with AdamW optimizer and use a grid search to tune the learning rate and batch size.

**SPDF on SQuADv1.1 Results**   We evaluate BERT Small with Sparse Wide, Sparse Parallel, and Sparse Factorized members of the Sparse-IFT family. All transformations, except Sparse Parallel,

Table 18: Size and architecture of the BERT Small model, which is trained using the setup from Nvidia [66]

| Model | $n_{params}$ | $n_{layers}$ | $d_{model}$ | $n_{heads}$ | $d_{head}$ |
|---|---|---|---|---|---|
| BERT Small | 29.1M | 4 | 512 | 8 | 64 |

perform comparably to the dense baseline on SQuAD. Unlike CV architectures, BERT initializes the layers with a normal distribution, which has an adverse effect when layers undergo shape transformations (e.g., changes in depth [96], or width [92]). In our initial experiments, we found changing the initialization of BERT enables other families to outperform the dense baseline. In addition to initialization, BERT training has over six hyperparameters. We leave optimizing and analyzing the effect of these hyperparameters on Sparse-IFT for future work and restrict our current scope to demonstrating gains without tuning any hyperparameters. Using the Sparse Parallel IFT with 50% sparsity leads to a 0.7% improvement in the exact match (EM) accuracy over the dense baseline (see Table 19).

Table 19: Evaluation of Sparse Parallel IFT for pre-training BERT Small. We report EM (higher is better) obtained by sparse fine-tuning and dense fine-tuning BERT models on SQuADv1.1, respectively.

| Dense | Transformation | Fine-Tuning Method | 0.50 | 0.75 |
|---|---|---|---|---|
| 70.6 | Sparse Parallel | Sparse | **70.7** | 69.9 |
| | | Dense | **71.3** | 70.8 |

### C.2.2 SPDF on GPT

**Pre-training Experimental Setup** Here, we pre-train the models on the Pile [23] dataset. To train all GPT models, we use AdamW optimizer [57] with $\beta_1 = 0.9$, $\beta_2 = 0.999$ and $\epsilon = 10^{-8}$. The global norm is clipped at 1.0, and a weight decay of 0.1 is used. There is a learning rate warmup over the first 375M tokens, followed by a cosine decay to 10% of the peak learning rate. We follow the recently published Chinchilla [33] recommendations for obtaining loss-optimal pre-trained baseline configurations of models. The context window size is 2048 following [2]. Table 20 shows a detailed breakdown of the model architectures, learning rate, and training settings. In Table 16, we outline the architecture configurations for Sparse Wide IFT 50% and 75% variants.

Table 20: Size, architecture, and learning hyperparameters (batch size and learning rate) of the GPT-3 Small model, which is trained using Chinchilla optimal configurations ($\approx$ 20 tokens per parameter)

| Model | $n_{params}$ | $n_{layers}$ | $d_{model}$ | $n_{heads}$ | $d_{head}$ | Batch Size | Learning Rate | Training Tokens |
|---|---|---|---|---|---|---|---|---|
| GPT-3 Small | 125M | 12 | 768 | 12 | 64 | 256 | $6\times10^{-4}$ | 2.5B |

**Fine-tuning Experimental Setup** We finetune the Sparse Wide IFT variants of GPT-3 Small on the WikiText-103 [60] dataset following the setup presented in [71]. We finetune for ten epochs and perform early stopping once the models overfit. We performed a grid search to discover an appropriate learning rate that led to the best perplexity for a given compute budget. More specifically, on the dense baseline and Sparse Wide IFT variants, we use a batch size of 32 and select the best learning rate among {5e-3, 3e-3, 1e-3, 3e-4, 1e-4, 3e-5, 1e-5} on the validation set.

In Tables 16, 18, and 20, $n_{params}$ is the total number of trainable parameters, $n_{layers}$ is the number of decoder layers, and $d_{model}$ is the base size of the model. The feedforward bottleneck is four times the base size, i.e., $d_{ff} = 4 \times d_{model}$. Finally, $n_{heads}$ is the number of attention heads, and $d_{head}$ is the dimension of each attention head.

**SPDF on WikiText-103 Results** Here, we pre-train a GPT-3 Small architecture with Sparse Wide IFTs at 50% and 75% sparsity. Post pre-training, we finetune our models on WikiText-103. The GPT-3 Small 75% Sparse Wide model reduces the perplexity (PPL) by a noticeable 1.3 points compared to dense (refer to Table 21).

Table 21: Evaluation of Sparse Wide IFT for pre-training GPT-3 Small. We report perplexity (lower is better) obtained by sparse fine-tuning and dense fine-tuning GPT models on Wikitext-103, respectively.

| Dense | Transformation | Fine-Tuning Method | 0.50 | 0.75 |
|---|---|---|---|---|
| 15.9 | Sparse Wide | Sparse | **15.6** | 16.0 |
| | | Dense | 15.1 | **14.6** |

# D Benchmarking Efficiency w.r.t Model Parameters and Wall-clock

In this section we evaluate the training and inference efficiency of Sparse-IFT with respect to number of model parameters and wall-clock time.

## D.1 Model Parameters

In this section, we evaluate Sparse-IFT variants using test-accuracy w.r.t model parameters as our metric. Figure 3 compares the performance of Sparse Wide IFT variants of ResNet on ImageNet dataset. As shown in the figure, Sparse Wide IFT variants obtain significantly higher accuracy compared to the dense counterparts. This result indicates that Sparse-IFT variants are also efficient for inference setup, as they obtain higher accuracy for a fixed amount of parameter/storage budget compared to dense models.

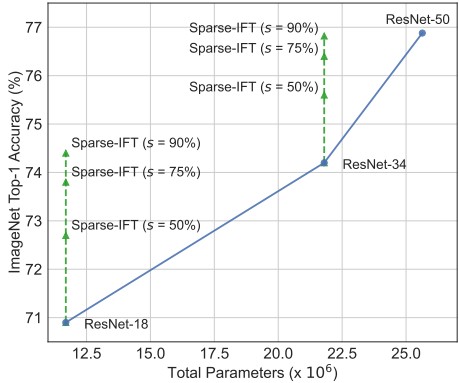

Figure 3: Accuracy vs number of parameters for different variants of ResNet on ImageNet. Sparse-IFT provides significant accuracy gains across different models and sparsity levels. In particular, the best Sparse-IFT variants of ResNet-18 and ResNet-34 achieve 3.5% and 2.7% improvements over their dense baselines, respectively.

## D.2 Wall-clock Time

Results presented in Section 3 validate our hypothesis, i.e., training DNNs with dense matrices is FLOP inefficient. Replacing dense layers with Sparse-IFT increases the training efficiency by providing significantly higher accuracy using the same amount of training FLOPS. This result is significant from a theoretical perspective but does not translate to direct practical value on hardware that can not accelerate unstructured sparsity (e.g., Nvidia GPUs, Google TPUs). However, there has recently been a renewed interest in hardware software co-design for accelerating unstructured sparsity. Here, we benchmark Sparse-IFT on these platforms to demonstrate its practical value. We hope these results motivate the broader machine learning community to explore and exploit the benefits of unstructured sparsity for training and inference.

**Inference Setup** We use Neural Magic's DeepSparse [36, 43] tool for benchmarking Sparse-IFT variants. The benchmarking is conducted on G4dn instances available on the AWS cloud. These instances support the AVX-512 instruction set, which is used by the DeepSparse inference runtime

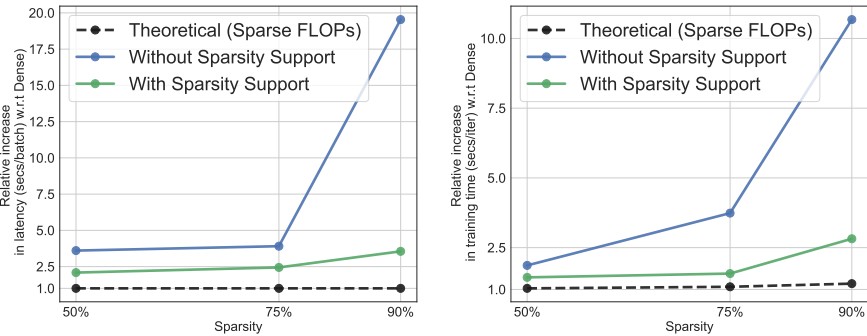

Figure 4: Benchmarking (left) inference on Neural Magic's DeepSparse runtime and (right) training acceleration with unstructured sparsity on the Cerebras CS-2. In both setups, we measure the relative increase in latency or training speed for Sparse-IFT variants against the dense model.

to accelerate unstructured sparsity. We benchmark different configurations of the Sparse Wide ResNet-18 model with sparsity $\in \{50\%, 75\%, 90\%\}$ for batched inference on ImageNet. We report runtime for batch-inference of 64 images at $224 \times 224$ resolution.

**Training Setup** We evaluate the training efficiency of Sparse-IFT on the Cerebras CS-2 which supports and accelerates training with unstructured sparsity (both forward and backward passes). We benchmark the training speed measured in seconds/iteration. Note that the overall FLOPs of models in the GPT family are comprised of matrix multiplication FLOPs and attention FLOPs. Attention FLOPs (i.e., spent in multi-head attention) scale quadratically with sequence length and are invariant to weight sparsity. To demonstrate the efficacy of sparse kernels for unstructured weight sparsity, we report our results for dense and Sparse Wide variants of the GPT-3 1.3B model with a sequence length of 256 and batch size of 528. We benchmark different configurations of Sparse Wide GPT-3 1.3B with sparsity $\in \{50\%, 75\%, 90\%\}$ and report seconds/ iteration.

Figure 4 shows the result of benchmarking inference and training of Sparse-IFT Sparse Wide family. In both setups, we measure the relative increase in latency or training speed for Sparse-IFT variants against the dense model. Note that configurations of Sparse-IFT at different values of sparsity do not incur a significant change in the FLOPs compared to the dense model. On ideal hardware, FLOPs should translate directly to wall clock time, and hence, the inference latency or training time for all configurations of Sparse-IFT should be the same as that of the dense model (dotted black line). Conversely, when hardware does not support unstructured sparsity, the latency or training time of Sparse-IFT variants increases with sparsity (blue line). Our results lie between these two spectrums (green line). Using Neural Magic's sparse inference runtime, we observe a significant reduction in inference latency, bringing down the relative increase in latency from 19.5x to 3.5x. Similiarly, in the case of training on the Cerebras CS-2, we observe a significant reduction in training-time, bringing down the relative increase from 10.6x to 2.8x.

## E Author Contributions

We provide a summary of each author's contributions:

- Vithursan Thangarasa was an integral part of the project by participating in discussions with Shreyas Saxena and contributing to the method. He also implemented all Sparse-IFT transformations in PyTorch, proposed using non-linearity in Sparse-IFT, conducted experiments for the entire study on CIFAR-100 and its ablations, obtained initial results on ImageNet, extended Sparse-IFT to efficient architectures (e.g., BotNet, MobileViT), conducted the entire study with GPT on Cerebras CS-2, and contributed to writing parts of the manuscript.

- Shreyas Saxena conceived the key idea of matching the FLOPs of Sparse Wide transformation to a compact dense model, extended the idea to other members of the Sparse-IFT family, helped with the implementation, established cardinality of Sparse-IFT members to explain

the results, conducted experiments for BERT, benchmarked Sparse-IFT for inference, and wrote majority of the manuscript.

- Abhay Gupta validated sparse optimizers in PyTorch, conducted experiments with Sparse-IFT ResNet variants on ImageNet, obtained results with MobileNetV2 architecture, helped with pre-training of Sparse-IFT variants of GPT on Cerebras CS-2, conducted all experiments of Sparse-IFT on downstream CV tasks, and contributed to writing parts of the manuscript.

- Sean helped with the bring-up of sparsity support on Cerebras CS-2 which was crucial for benchmarking and training Sparse-IFT variants of GPT models, and provided feedback to improve the structuring and presentation of the manuscript.

