# OpenReview forum: "Sparse Iso-FLOP Transformations for Maximizing Training Efficiency"
_NeurIPS.cc/2023/Workshop/WANT — WANT@NeurIPS 2023 Poster_

### Official Review · Reviewer_imbu · 2023-10-24
**This paper suggests a novel method for increasing training efficiency by performing sparsity-friendly replacements in model. Work has solid theoretical and strong empirical evidence of proposed improvements. My recommendation: ACCEPT.**

**Confidence:** 5

**Review:**

## Summary

This paper suggests a novel method for increasing training efficiency by performing sparsity-friendly replacements in the model. Work has solid theoretical and strong empirical evidence of proposed improvements. Authors perform experiments on a wide range of CNN and Transformer-based architectures. An extensive ablation study is conducted with cohesive conclusions.

## Strengths

1. Writing is easy to follow, language is good
2. High-quality self-explanatory visual elements
3. Extensive details for results reproduction
4. Deep ablation study with cohesive conclusions


## Weaknesses

1. **Reasons for latency and training time increase even for hardware supporting unstructured sparsity are unclear**

   Mainly referring to Figure D.4. It is clear that due to lack of hardware support latency and training time grow significantly. However, it is not clear why latency and training time increase even for hardware supporting unstructured sparsity. In the paper, it is stated that the number of FLOPs is kept the same before and after applying transformations. Even more, hardware accelerators with sparse computations supported report computation acceleration compared to dense computations. Please explain the reasons.

2. **Programming code is absent**

   Thus, it complicates the reproduction of results, but all necessary details like reference training pipelines, hardware setup, and pre-processing logic are well described. It is highly recommended to make the code public due to its high practical value.

3. **Text imperfections**

   - High volume: 9 pages of main text
   - Too big references list: 5 pages (!)
   - Unobvious tables highlights Within tables it is not clear how it is highlighted: row-based or column-based.

---

### Official Review · Reviewer_RaEm · 2023-10-26
**Interesting method, new way to look at the lottery ticket hypothesis. But the applicability could be limited due to special hardware necessary to achieve effective sparse x dense multiplication.**

**Confidence:** 3

**Review:**

# Quality and Clarity
## Pros
- The method explores an interesting implementation of a well-known lottery-ticket hypothesis.
- Description of all types of Sparse Iso-FLOP transformations is comprehensive, backed up by FLOPs estimations.
- Detailed ablation studies on comparison between sparse training regimes (static, SET, RigL), types of Iso-FLOP transformations, etc.
- Results for ResNet family of models shows great accuracy increase on ImageNet dataset. Results on IFT GPT3-small shows lower perplexity, than the original.
- Benchmarking of inference and training steps on platforms supporting unstructured sparsity in Appendix D.

## Cons
- There are a few typos throughout the text (i.e. line 370).
- FLOPs estimation for sparse x dense matrix multiplications provided in the manuscript can only be achieved on a special hardware (i.e. Cerebras CS-2) since traditional GPU architectures are only efficient in dense operations(lines 866-867). At the same time this special hardware doesn’t support dynamic sparsity training (as mentioned in lines 211-212), with which most of results in the paper were reported (except for results on GPT3, as mentioned in lines 208-210). Given that, accuracy results reported for ResNet models were achieved by training a higher FLOPs model than the original (k^2 time FLOPs of usual dense multiplication for Sparse Wide layers) which is not entirely clear until the Appendix section and should be emphasised earlier in the manuscript.
- No comparison to state-of-the-art distillation/pruning methods, which achieve the same FLOPs by reducing parameters of larger models.

# Originality and Significance
Interesting method, new way to look at the lottery ticket hypothesis. But the applicability could be limited due to special hardware necessary to achieve effective sparse x dense multiplication. Training IFT networks without this hardware can quickly become unfeasible (as was the case with GPT3). And currently, this hardware does not support dynamic sparsity training which produced the accuracy results for IFT ResNet family model training.

---

### Official Review · Reviewer_gKzm · 2023-10-27
**Really impressive results in vision setting needs a bit of work for intuition and analysis.**

**Confidence:** 3

**Review:**

This paper proposes a new pre-training strategy where they apply sparse ISo-flops transformations (unstructured sparsity) during pre-training from scratch and achieves higher performance (acc/ppl) using the same amount of training flops. The paper has shown impressive improvements in vision domain both in Imagenet 1K top acc and finetuning performance using MScoco and city spaces. The paper is well written and idea is clearly conveyed. The technique seems novel. Some key areas of improvement of this work are as follows/
1) The core intuition of why this technique works is written in sec 2.4 but it is still unclear to me why it works. IF higher width or depth is the key then why baseline wider or deeper NN achieves lesser accuracy? Why more sparsity leads to higher accuracy with the same width (table 2).?
2) The PPL results in Table 9 for GPT-3 baseline looks unusually high. I am curious why this is the case?
3) The technique has worked really well on vision model where the training is done for multiple epochs but miserably failed in NLP setting. Moreover the pattern of higher sparsity leads to higher accuracy in vision settings changed in NLP setting. Any thoughts why?
It may be due to overfitting issue in vision (caused by multi epoch training) but I am not fully sure.

Overall the results are interesting and in my opinion this paper should be accepted (borderline).

---

### Meta-Review · Area_Chair_QcRb · 2023-10-27

**Recommendation:** Accept (Poster)
**Confidence:** 4

**Metareview:**

**Strengths:**
* Reviewers found the paper to be well-written, with the core concepts clearly explained.
* Novel contribution.
* Strong evaluation with detailed ablation studies, although comparisons to related pruning/distillation work appears to be missing.

**Weaknesses:**
* FLOPs may not correspond to direct speedups - reviewers point out that realizing hardware speedups with unstructured sparsity is challenging on common parallel architectures such as GPUs.
* Relatively weaker results on NLP tasks when compared to vision benchmarks.

This appears to be a borderline paper based on reviews.

---

### Decision · Program_Chairs · 2023-10-28

**Decision:**

Accept (Poster)

**Comment:**

We thank the authors for their time and contribution to WANT and we are pleased to share that after the reviewing process the paper has been accepted. Congratulations! We encourage the authors to consider reviewers' feedback for the improvement of the camera-ready version. We hope to see you in person at the workshop and brainstorm on efficient training research together!